A pyroptosis gene-based prognostic model for predicting survival in low-grade glioma

Wang Hua 1 2
Yan Lin 3
Liu Lixiao 4
Lu Xianghe 1 2
Chen Yingyu 1
Zhang Qian 1
Chen Mengyu 1
Cai Lin longcailin1021@163.com 5
Dai Zhang’an vivian19941204@163.com 1 2
1 Department of Neurosurgery, The First Affiliated Hospital of Wenzhou Medical University , Wenzhou , Zhejiang , China
2 Zhejiang Provincial Key Laboratory of Aging and Neurological Disorder Research, The First Affiliated Hospital of Wenzhou Medical University , Wenzhou , Zhejiang , China
3 Department of Breast Surgery, The Second Affiliated Hospital and Yuying Children’s Hospital of Wenzhou Medical University , Wenzhou , Zhejiang , China
4 Department of Gynaecology and Obstetrics, The First Affiliated Hospital of Ningbo University , Ningbo , Zhejiang , China
5 Department of Neurosurgery, Shanghai Sixth People’s Hospital Affiliated to Shanghai Jiao Tong University School of Medicine , Shanghai , China
Date Swapneeta
Electronic publication date: 2023 Nov 13
Publication date: 2023
Volume: 11
Electronic Location ID: e16412
Received 2023 May 24; Accepted 2023 Oct 15
Copyright: ©2023 Wang et al.
Copyright year: 2023
Copyright holder: Wang et al.
License: This is an open access article distributed under the terms of the Creative Commons Attribution License, which permits unrestricted use, distribution, reproduction and adaptation in any medium and for any purpose provided that it is properly attributed. For attribution, the original author(s), title, publication source (PeerJ) and either DOI or URL of the article must be cited.
License URL: https://creativecommons.org/licenses/by/4.0/

Keywords: Pyroptosis, Low-grade gliomas, Risk scoring model, Prognosis, Individualized treatment

Funding: Wenzhou Municipal Science and Technology Bureau of China Y2020063 This work was supported by the Wenzhou Municipal Science and Technology Bureau of China (grant number Y2020063). The funders had no role in study design, data collection and analysis, decision to publish, or preparation of the manuscript.

==============================
Background

Pyroptosis, a lytic form of programmed cell death initiated by inflammasomes, has been reported to be closely associated with tumor proliferation, invasion and metastasis. However, the roles of pyroptosis genes (PGs) in low-grade glioma (LGG) remain unclear.

Methods

We obtained information for 1,681 samples, including the mRNA expression profiles of LGGs and normal brain tissues and the relevant corresponding clinical information from two public datasets, TCGA and GTEx, and identified 45 differentially expressed pyroptosis genes (DEPGs). Among these DEPGs, nine hub pyroptosis genes (HPGs) were identified and used to construct a genetic risk scoring model. A total of 476 patients, selected as the training group, were divided into low-risk and high-risk groups according to the risk score. The area under the curve (AUC) values of the receiver operating characteristic (ROC) curves verified the accuracy of the model, and a nomogram combining the risk score and clinicopathological characteristics was used to predict the overall survival (OS) of LGG patients. In addition, a cohort from the Gene Expression Omnibus (GEO) database was selected as a validation group to verify the stability of the model. qRT-PCR was used to analyze the gene expression levels of nine HPGs in paracancerous and tumor tissues from 10 LGG patients.

Results

Survival analysis showed that, compared with patients in the low-risk group, patients in the high-risk group had a poorer prognosis. A risk score model combining PG expression levels with clinical features was considered an independent risk factor. Gene Ontology (GO) and Kyoto Encyclopedia of Genes and Genomes (KEGG) analyses indicated that immune-related genes were enriched among the DEPGs and that immune activity was increased in the high-risk group.

Conclusion

In summary, we successfully constructed a model to predict the prognosis of LGG patients, which will help to promote individualized treatment and provide potential new targets for immunotherapy.

Introduction

Gliomas are neuroepithelial tumors originating from the supporting glial cells of the central nervous system (CNS). Glioma is a common malignant tumor of the central nervous system, accounting for more than 80% of all malignant tumors of the brain, and is characterized by a high incidence, high recurrence rate, high mortality rate and low cure rate (Huang et al., 2019; Sathornsumetee, Rich & Reardon, 2007). Gliomas can originate from astrocytic, oligodendroglial, mixed oligoastrocytic, or neuronal-glial cells; thus, there are a variety of pathological types of gliomas, including astrocytomas, oligodendrogliomas, mixed oligoastrocytomas, and mixed glioneuronal tumors (Forst et al., 2014). Gliomas are divided into 4 grades, grade 1 (lowest grade) through grade 4 (highest grade), based on the histopathological characteristics of the World Health Organization (WHO) classification system (Louis et al., 2016). According to the WHO grading system, LGG is considered to consist of diffuse low-grade glioma and intermediate-grade glioma (WHO grade II and III), which are mainly astrocytomas, oligodendrogliomas and mixed oligoastrocytomas (Brat et al., 2015), and high-grade glioma (HGG) is considered to consist of glioblastoma (GBM, WHO grade IV) (Rathore et al., 2020). Surgery, radiotherapy, chemotherapy or combination therapies and individualized management beginning as soon as possible based on tumor location, histology, molecular characteristics, and patient characteristics are the main treatment methods currently used to treat glioma patients (Forst et al., 2014). However, patients with glioma still suffer from high mortality, mainly due to insufficient understanding of the molecular pathogenesis of the disease and a lack of timely diagnosis and tools for sensitive treatment monitoring (Aili et al., 2021). Therefore, we urgently need to find biological targets and build predictive models to reduce the mortality of glioma patients.

At present, an increasing number of studies are underway to find molecular biological targets of gliomas and construct genetic risk scoring models for prompting personalized treatment. However, current research focuses more on the finding of biological targets and the construction of predictive models for HGG patients, rather than LGG patients, which has led to a scarcity of molecular biological targets for the individualized treatment of LGG patients. Zhu et al. (2021) constructed a ferroptosis-related gene risk scoring model through three hub genes, which could be used to independently predict the prognosis of GBM patients. Subsequently, a model for accurately predicting the prognosis of GBM patients was successfully constructed based on the characteristics of nine glycolysis-related genes (Bingxiang et al., 2021). In addition, based on five prognosis-related genes identified in a screen, (Lei et al., 2021) constructed a metabolic gene-related risk score model to predict the prognosis of GBM patients. Despite certain advances in the study of GBM, the life expectancy of patients with GBM has not been obviously improved (Gilbert et al., 2013; Kaur et al., 2020). Studies have shown that the five-year survival rate and median survival of LGG patients are significantly longer than those of HGG patients (Claus et al., 2015; Lei et al., 2021). In addition, recent studies have shown that early diagnosis and reasonable treatment can significantly improve the survival rate of LGG patients (Liu et al., 2021). Therefore, it is very important to promote the early diagnosis of LGG with diagnostic biological indicators to further improve the survival of LGG patients. Although studies have shown that TP53, EGFR and MGMT promoter methylation are considered as prognostic factors for LGG in clinical practice, these cannot be targeted to assess the survival outcome of LGG patients accurately (Liu et al., 2012; Sun et al., 2014; Weller et al., 2012). Hence, there is an urgent need for the identification of new biological indicators for diagnosis and prognosis in LGG patients.

In the present study, we aimed to find biological targets and construct predictive models for LGG patients. After screening, we finally obtained nine HPGs and constructed a genetic risk score model based on these genes. The expression levels of the nine HPGs were verified in normal brain tissue and tumor tissue of 10 LGG patients. Finally, we found that the model can be used to predict the survival risk of LGG patients. These results suggest that the factors identified in this study can be used for targeted therapy and individualized treatment of LGG patients.

Materials and Methods

Data collection

Gene expression profiles and corresponding clinical information from 529 LGG patients were collected from The Cancer Genome Atlas database (TCGA, https://portal.gdc.cancer.gov/). We excluded 53 samples with very short survival times (OS<60 days) and used the remaining 476 samples for the subsequent analysis. Due to the lack of relevant data on normal brain tissue in the TCGA database, data from 1,152 normal brain tissue samples were downloaded from the Genotype–Tissue Expression database (GTEx, https://commonfund.nih.gov/gtex) and served as a normal control group. In addition, the gene expression profiles and corresponding clinical information for 50 LGG patients in the GSE43378 dataset from the Gene Expression Omnibus database (GEO, https://www.ncbi.nlm.nih.gov/geo/) were used for independent validation.

Acquiring PGs

We searched for “Pyroptosis” in the Gene Set Enrichment Analysis (GSEA, http://www.gsea-msigdb.org/gsea/index.jsp) database and downloaded gene sets associated with PGs, which allowed us to obtain 27 PGs. We combined this list with the list of 51 PGs in a previous article (Zhang et al., 2021) to yield a total of 51 PGs.

Identification of DEPGs

Tumor tissue samples from the TCGA database and normal tissue samples from the GTEx database were integrated for subsequent research. Thereafter, we used the R package “limma” to perform variation analysis with the integrated gene expression. P < 0.05 were the essential screening criteria for DEPGs. The DEPGs were then visualized by using the “ggplot2” package with parameters of P < 0.05 for the volcano map and the “pheatmap” package with cut P = 0.05 for the heatmap.

Functional enrichment analysis

To further explore the DEPGs, the R package “clusterProfiler” was used for gene function enrichment analysis. Gene Ontology (GO) and Kyoto Encyclopedia of Genes and Genomes (KEGG) enrichment analyses were used to reveal the functions of the DEPGs and their related pathways. GO enrichment analysis was used to explore the biological processes (BP), molecular functions (MF) and cellular components (CC) of DEPGs. KEGG pathway analysis was used to analyze the signaling pathways of the DEPGs.

Construction of a PG risk score model

To further explore the prognosis-related genes in the DEPGs, we used the R package “survival” to perform univariate Cox regression analysis on the TCGA training set (OS ≥ 60 days) to identify prognosis-related genes. In addition, P < 0.05 was chosen as the filter to select significantly associated genes for subsequent model construction. Next, the mRNAs identified in the Cox regression results were subjected to LASSO regression dimensionality reduction, mainly relying on the R package “glmnet”. To construct a further accurate regression model, lambda screening with cross-validation was conducted.

Finally, we used multivariate Cox regression to identify the most relevant hub genes for LGG prognosis in DEPGs and constructed a predictive prognostic genetic risk score model. The sum of the products of the expression of the individual genes and the corresponding coefficients was used to calculate the risk score. Kaplan–Meier (KM) curves were constructed using the R packages “tidyverse” and “survminer” to determine the difference in OS between the high- and low-risk groups. ROC curves were constructed using the R package “timeROC” to assess the predictive accuracy of the genetic risk score model.

Analysis of genetic changes

Gene mutation analysis of nine HPGs was performed using the cBioPortal database (http://www.cbioportal.org). The cBioPortal database can be used to analyze the type and frequency of alterations in genes, including copy number amplifications, deep deletions and missense mutations of unknown significance.

Construction and validation of a prognostic nomogram for OS

Univariate and multivariate Cox analyses were performed on clinicopathological parameters (including age, grade and stage) and risk scores to assess whether the risk score model was independent of each parameter. Independent prognostic factors identified by multivariate Cox regression were combined to construct a nomogram using the R package “rms” and the R package “survivor” to predict 3-, 5- and 7-year survival. Moreover, calibration curves were used to validate their predictive values.

Immune cell correlation analysis

Immune cell infiltration tables for TCGA tumors were downloaded through TIMER2.0 (http://timer.comp-genomics.org/). Risk scores and immune cell correlations were analyzed using the R packages “limma”, “scales”, “ggplot2”, “ggtext”, and “ggpubr”. Immunological differences between the high- and low-risk groups were compared based on the results of the analysis.

External validation of the risk score model

To validate the predictive prognostic value of the risk score model, we used the GSE43378 dataset from the GEO database as an external validation group. Risk scores were calculated using the formula developed using the training group. KM curves were plotted to compare the OS between the two groups, and ROC curves were plotted to assess the stability of the model.

Establishment of the protein–protein interaction (PPI) network

To understand the interactions between DEPGs, we constructed a PPI network with the open-source software platform STRING (https://string-db.org/) to build the network model. Gene scores >0.4 and P < 0.05 were the essential screening criteria for the network model.

Tumor and paracancerous tissue collection from LGG patients

Our study was approved by the Clinical Research Ethics Committee of The First Affiliated Hospital of Wenzhou Medical University (permission: 2016-076), and written informed consent was obtained from all patients. The study protocol was carried out according to the principles of the Helsinki Declaration. Ten LGG patients (Grade II, n = 5; Grade III, n = 5) who had not received preoperative radiotherapy or chemotherapy underwent surgery between 2019 and 2021 at the First Affiliated Hospital of Wenzhou Medical University. The samples in this study were from five males and five females, aged between 26 and 62 years (mean, 41.5 years). The clinical characteristics of the patients are shown in Table 1.

Table 1 Clinicopathological characteristics of 10 LGG patients.

Patientc number	Gender	Grade	Age	Treatment method	Treatment after surgery	Max diameter (cm)	MGMT	IDH1	
1	F	3	33	Sur	T+R	7.1	+	+	
2	F	3	62	Sur	T+R	4	–	–	
3	M	3	26	Sur	T+R	4	+	+	
4	M	3	57	Sur	T+R	4.2	–	–	
5	M	2	47	Sur	T+R	6.5	+	+	
6	F	3	31	Sur	T+R	5	–	+	
7	F	2	48	Sur	R	3.5	–	+	
8	M	2	36	Sur	T	6.7	+	+	
9	M	2	34	Sur	R	7.5	–	+	
10	F	2	41	Sur	T	5	+	–	
Notes.

F, Female; M, Male.

Sur, Surgery; Un, no treatment; T, temozolomide; R, radiotherapy.

MGMT, O-6-methylguanine-DNA methyltransferase; IDH1, isocitrate dehydrogenase 1.

Detection of the gene expression levels of nine HPGs in tumors and paracancerous tissues

Total RNA was extracted from the tumor tissue and paracancerous tissue using TRIzol reagent (Thermo Fisher Scientific, Waltham, MA, United States) according to the manufacturer’s instructions. Single-stranded cDNA was synthesized from 1 mg of total RNA using the PrimeScript RT Reagent Kit with gDNA Eraser (Takara Biotechnology Co., Ltd., Dalian, China). Each cDNA sample (2 µl) was amplified in SYBR Green Real-time PCR Master Mix (final volume, 20 µL, YEASEN, China, #11203ES03), and the mRNA expression of the hub genes in the amplified samples was detected by a 7500 PCR system (Thermo Fisher Scientific). Thermal cycling conditions were as follows: 95 °C for 5 min, followed by 40 cycles of 95 °C for 20 s and 60 °C for 30 s. qPCR assays were conducted in triplicate in a 10 mL reaction volume for each sample. The relative expression of mRNA was calculated by the 2- ΔCt method. The primers used are shown in Table 2.

Table 2 The primers used to amplify the nine HPGs.

Primer name	Forward sequence	Reverse sequence	
CASP4	CCTCTGACAGCACATTCTTGGTACTC	CAGTTGCGGTTGTTGAATATCTGGAAG	
TP63	GCAACGCCCTCACTCCTACAAC	AGTCCATTCATGTCTCCAGCCATTG	
TIRAP	CTAGGCAAGATGGCTGACTGGTTC	GGGTAGTGGGCTGTCCTGTGAG	
CASP9	GACCAGAGATTCGCAAACCAGAGG	AAGAGCACCGACATCACCAAATCC	
IL18	TGCCCTCAATCCCAGCTACTCAAG	TCTCGGCTCACCACAACCTCTAC	
TNF	AAGGACACCATGAGCACTGAAAGC	AGGAAGGAGAAGAGGCTGAGGAAC	
IL1A	GACCAACCAGTGCTGCTGAAGG	AATAGTTCTTAGTGCCGTGAGTTTCCC	
PLCG1	CTGGATGTTGCTGCCGACTCAC	CTTCCTCCGTTCCATTATCTTCCCTTC	
TP53	GCCCATCCTCACCATCATCACAC	GCACAAACACGCACCTCAAAGCAGC	

Statistical analysis

All analyses in this study were performed using R version 4.1.1 (R Core Team, 2021) and GraphPad Prism (version 8.3; GraphPad Software, La Jolla, CA, USA). The data between the two groups were compared with Student’s t test. Unless otherwise specified, P < 0.05 was considered significant.

Results

Clinical characteristics of the patient sample

The analysis process of our study is shown in Fig. 1. A total of 476 LGG patients in the TCGA dataset met the criteria for the training group. Fifty LGG patients in the GSE43378 dataset were used for the validation group. The detailed clinical characteristics of the training group are shown in Table 3.

Figure 1 Flow diagram of the study selection process.

Workflow diagram for building a risk assessment model for PGs in LGG.

Table 3 Clinicopathological characteristics of LGG patients in the training group.

Characteristics	Number	Percentage (%)	
Age			
≤ 65	449	94.32	
>65	27	5.68	
Sex			
Female	214	44.96	
Male	262	55.04	
Survival			
No	122	25.63	
Yes	354	74.37	
Survival time			
< 1 year	82	17.23	
≥1 and < 3 years	237	49.79	
≥3 and < 5 years	92	19.33	
≥5 and < 7 years	25	5.25	
≥7 years	40	8.40	

Identification of DEPGs in LGG

We screened 45 DEPGs between tumor samples (TCGA) and normal samples (GTEx) by comparing the expression levels of 51 PGs. Among them, 25 genes (CHMP6, NOD2, NLRC4, CHMP4B, CASP3, BAX, TIRAP, CASP9, BAK1, NLRP6, IL18, NLRP3, CASP5, GSDMC, TNF, IL1A, CASP1, AIM2, PYCARD, CHMP7, CHMP2B, CASP6, IL1B, TP53 and IRF2) were upregulated, while the remaining 20 genes (CASP4, NLRP1, TP63, CHMP4C, SCAF11, CYCS, HMGB1, NOD1, PRKACA, CHMP2A, GPX4, CASP8, NLRP2, ELANE, GSDMD, PLCG1, GZMB, IL6, IRF1 and GSDMB) were downregulated, as illustrated by heatmap (Fig. 2A) and volcano map (Fig. 2B). In order to further explore the signal pathway and interrelationship of 45 DEPGs, KEGG analysis showed that 45 DEPGs were mainly enriched in the NOD—like receptor signaling pathway, Salmonella infection and Necroptosis (Fig. 2C).

Figure 2 The expression and interrelationship of DEPGs.

(A) Heatmap (green: low expression level; red: high expression level) showing the expression of the DEPGs between the normal (N, blue) tissues and the tumor tissues (T, red). (B) Volcano map showing DEPG expression. Upregulated genes are represented in red, and downregulated genes are represented in purple. (C) Barplot graph for KEGG pathways with 45 DEPGs.

In summary, 45 differentially expressed pyroptosis genes were selected for subsequent analysis.

Construction of a risk score model to predict patient prognosis

To further screen for DEPGs associated with LGG prognosis, we analyzed the relationship between the expression levels of 45 DEPGs and the OS of LGG patients in the training group by univariate Cox regression. The results showed that the expression levels of 29 prognostic genes were significantly associated with OS in 476 LGG patients. Among them, five prognostic genes (HR <1) were protective genes, while the remaining 24 prognostic genes (HR > 1) were associated with poorer outcome in LGG patients (P < 0.05) (Fig. S1A). Subsequently, we reduced the number of candidate genes to 13 by LASSO analysis to obtain a better fitting model (Figs. S1B and S1C). Finally, the results of multivariate Cox regression analysis showed that nine hub genes associated with PGs, namely, CASP4, TP63, TIRAP, CASP9, IL18, TNF, IL1A, PLCG1, and TP53, could be used to construct a risk score model. The prognosis-related risk score was calculated as follows: risk score = (0.476 × Expression of CASP4) + (0.509 × Expression of TP63) + (−0.399 × Expression of TIRAP) + (−0.618 × Expression of CASP9) + (0.279 × Expression of IL18) + (−0.341 × Expression of TNF) + (0. 563 × Expression of IL1A) + (1.093 × Expression of PLCG1) + (0. 356 × Expression of TP53). As shown in Fig. 3A, the 476 LGG patients were divided into high-risk and low-risk groups based on the median risk score obtained from the formula. Survival analysis showed that patients in the low-risk group had a longer survival time and fewer deaths than those in the high-risk group (Fig. 3B). The expression of the nine HPGs in the high-risk and low-risk groups is shown in Fig. 3C. The expression levels of TIRAP, CASP9 and TNF were increased in the low-risk group compared to the high-risk group. In contrast, the expression levels of the remaining six HPGs were decreased in the low-risk group. The results of survival analysis showed that patients with high expression in TIRAP, CASP9 and TNF are significantly higher than the OS of patients with low expression (Figs. 3D–3F). The remaining genes show the opposite results (Figs. 3G–3L).

Figure 3 Construction of the risk score model.

(A) Distribution of patient risk scores. (B) Scatter plot depicting the survival status and survival time of patients with different scores. The red and blue dots represent death and survival, respectively. (C) A heatmap indicating the expression levels of nine HPGs in the high- and low-risk groups. (D–F) Compared with the high- expression group, patients with low- expression of CASP9, TIRAP and TNF had a poorer prognosis. (G–L) Compared with the high- expression group, patients with low- expression of PLCG1, IL18, TP63, TP53, CASP4 and IL1A had a better prognosis.

These results showed that we could predict the prognosis of LGG patients by calculating their risk score.

HPG expression level analysis

To further explore the expression levels of the nine HPGs, we analyzed data from 529 LGG patients in the TCGA dataset as well as 1,152 normal brain tissues in the GTEx. Comparing the expression levels of the nine HPGs between tumor samples in the TCGA dataset and normal samples in the GTEx dataset, we found that although CASP4, TP63, and PLCG1 were considered risk genes, the expression of these genes was elevated in normal brain tissue compared to the tumor group (Figs. 4A–4C) (P < 0.001). In contrast, the expression of TIRAP, CASP9, IL18, TNF, IL1A, and TP53 was reduced in normal brain tissues (Figs. 4D–4I) (P < 0.001). This conclusion was also confirmed by RT-qPCR results, as shown in Figs. 4J–4R. By analyzing the differential expression of RNA levels of these nine hub genes in the tumors of LGG patients and paired paracancerous tissues, we also found that compared with the tumor tissues, the expression of paracancerous tissues in LGG patients was significantly increased by 940.93 ± 235.41% at CASP4 (p < 0.001), 76.05 ±  21.42% at PLCG1 (p < 0.001), and 196.50 ± 29.66% at TP63 (p < 0.001) (Figs. 4J–4L). However, the expression of paracancerous tissues in LGG patients was significantly reduced by 271.33 ± 52.07% at CASP9 (p < 0.001), 108.95 ± 20.35% at IL1A (p < 0.001), and 1,292.90 ± 527.72% at IL18 (p < 0.001), 2,496.74 ±  519.85% at TIRAP (p < 0.001), 9,446.02 ± 1,240.85% at TNF (p < 0.001), 6,240.67 ± 1387.22% at TP53 (p < 0.001), compared with the tumor tissues (Figs. 4M–4R).

Figure 4 The expression levels of nine HPGs in the tumor and normal groups.

(A–C) Compared with that in the tumor group, the expression of CASP4, PLCG1, and TP63 was significantly increased in the normal group. (D–I) Compared with that in the tumor group, the expression of CASP9, IL1A, IL18, TIRAP, TNF and TP53 was significantly decreased in the normal group. ***(P < 0.001) Tumor group vs. Normal group. (J–L) Compared with those in the tumor group, the mRNA expression levels of CASP4, PLCG1, and TP63 were significantly increased in the paracancer group. (M–R) Compared with those in the tumor group, the mRNA expression levels of CASP9, IL1A, IL18, TIRAP, TNF and TP53 were obviously decreased in the paracancer group. ***(P < 0.001) Tumor group vs. Paracancer group.

The above results showed that there was a significant difference in the expression of nine HPGS between the paracancerous tissues and tumor tissues, suggesting that these genes might be mutated in tumor tissues.

Analysis of the structural changes in HPGs

To further explore gene mutations, we analyzed nine HPGs in 283 LGG patients from the cBioPortal database. The results showed that 156 of 283 LGG patients (55%) exhibited significant changes in genetic sequence, with TP53 (52%) being the most frequently altered gene, followed by TIRAP (2.8%), TP63 (2.5%), PLCG1 (1.8%), CASP4 (1.4%), IL18 (1.4%), TNF (0.7%), CASP9 (0.4%), and IL1A (0.4%) (Fig. 5A). Interestingly, TP53 is the most commonly mutated gene in almost all tumors, especially in astrocytoma (Ohgaki & Kleihues, 2007; Synoradzki et al., 2021). In addition, we further analyzed the types of changes in the genes. As shown in Fig. 5B, 136 patients (48.06%) had mutations, 15 patients (5.3%) had multiple alterations, three patients (1.06%) had amplifications, and two patients (0.71%) had large-scale deletions. The domains altered within the hub genes also varied. In the TP63 gene, the R343L mutation appeared in the P53 DNA-binding domain (P53) structural domain, and the R487C and G635Vfs*69 mutations appeared outside the structural domain (Fig. 5C). In the IL18 gene, the L180F mutation appeared in the interleukin-1/18 (IL1/18) structural domain, and the T22M mutation appeared outside the conformational domain (Fig. 5D). In the PLCG1 gene, the E1163K and E1163_E1164dup mutations appeared in the C2 domain (C2), and the Y754C mutation appeared outside the conformational domain (Fig. 5E). In the TNF gene, there was only one mutation, a G224RE mutation in the structural domain of TNF (Fig. 5F). In addition, there were a large number of mutations in the TP53 gene. The most common mutations were R273C, R273H and R273L of the P53 DNA binding domain (P53), and mutations in the P53 tetramerization motif (P53_tetramer) and a small number of mutations outside the structural domain were also observed. In addition, the CNV status of nine HPGs has been shown in Fig. S2.

Figure 5 Genetic alteration analyses of nine HPGs in LGG.

(A) Summary of alterations in nine HPGs among 283 LGG patients. (B) Mutation frequency of selected genes in LGG. (C) Mutations in TP63. (D) Mutations in IL18. (E) Mutations in PLCG1. (F) Mutations in TNF. (G) Mutations in TP53.

These results demonstrated that nine HPGs are mutated in LGG patients, and among these HPGs, TP53 presents the highest mutation rate.

Validation of the risk score model for survival prediction

On the basis of the median risk score, LGG patients in GSE43378 (n = 50) were divided into high-risk and low-risk groups. Meanwhile, to further confirm the validity of the model for predicting OS in LGG patients, we plotted the Kaplan −Meier curve and ROC curve associated with the model. The results of survival analysis showed that the OS of the high-risk group was significantly lower than that of the low-risk group in both the training and validation groups (Figs. 6A, 6E). In addition, ROC curve analysis was used to assess the accuracy of the model predictions. The results showed that the AUC values in the training group for 1-, 2- and 3-year survival were 0.871, 0.856 and 0.858, respectively (Figs. 6B–6D). The AUC values in the validation group for 1-, 2- and 3-year survival were 0.653, 0.764 and 0.738, respectively (Figs. 6F–6H).

Figure 6 Validation of the risk score model.

(A) Kaplan–Meier (K-M) survival curves for the high-risk and low-risk groups in the training group (P = 3. 331e−16). (B) Receiver operating characteristic (ROC) curve in the training group for 1-year survival (AUC =0.871). (C) ROC curve in the training group for 2-year survival (AUC =0.856). (D) ROC curve in the training group for 3-year survival (AUC =0.858). (E) Kaplan–Meier (K-M) survival curves for the high-risk and low-risk groups in the testing group (P = 3.683e−3). (F) ROC curve in the testing group for 1-year survival (AUC =0.653). (G) ROC curve in the testing group for 2-year survival (AUC =0.764). (H) ROC curve in the testing group for 3-year survival (AUC =0.738).

Therefore, the results indicated that the model has good specificity and sensitivity for predicting LGG prognosis.

Evaluation of the risk score model as an independent prognostic factor for LGG

To verify whether the genetic risk score obtained by the model could be used as an independent prognostic factor, we performed univariate and multivariate Cox analyses on clinical factors such as risk score, age, tumor grade, and gender in the training set. The results of the univariate Cox analysis showed that risk score (P < 0.001), age (P < 0.001), and tumor grade (P < 0.001) were significantly associated with OS in LGG patients (Fig. 7A). In addition, the results of the multivariate Cox analysis also showed that risk score (P < 0.001), age (P < 0.001) and tumor grade (P < 0.001) were independent prognostic factors in patients with LGG (Fig. 7B). Thus, the risk score model is a reliable prognostic indicator independent of other clinicopathological parameters. To further clarify the independence of the risk score model, we performed stratified analysis for each indicator. We divided the patients into two age groups using 65 years as the basis of division and divided the patients into male and female groups according to sex. We also divided the patients into G2 and G3 groups (WHO grade II and III) according to pathological classification. The results showed that the survival curve was not affected by age, sex, or tumor grade, which further confirmed that the risk score model could be used to predict the prognosis of LGG patients (Fig. 7C).

Figure 7 Assessment of the risk score and clinical variables as independent prognostic indicators.

(A) Univariate Cox regression analysis was performed on clinical features and risk score in the training group. (B) Multivariate Cox regression analysis was performed on clinical features and risk score in the training group. (C) Stratification analysis assessing the predictive ability of the risk score model in different subgroups.

Based on the results obtained from the above analysis, we constructed a nomogram incorporating age, sex, tumor stage, and risk score to predict 3-, 5- and 7-year OS in LGG patients (Fig. 8A). In addition, 3-, 5- and 7-year survival were predicted with good accuracy using the nomogram calibration curves (Figs. 8B–8D). Thus, our model can be used to better predict the prognosis of LGG patients.

Figure 8 Nomograms for OS prediction for LGG patients.

(A) The nomogram was used to predict the OS of LGG patients at 1, 3, and 5 years. (B) Calibration curve analysis of the 3-year survival prediction accuracy of the nomogram. (C) Calibration curve analysis of the 5-year survival prediction accuracy of the nomogram. (D) Calibration curve analysis of the 7-year survival prediction accuracy of the nomogram.

HPG functional enrichment analysis

To further clarify the correlations between the nine HPGs in the model and signal path, gene enrichment analysis was used to further understand the potential functions of the HPGs. We investigated the molecular processes involved in LGG progression in depth by GO enrichment analysis and KEGG pathway analysis. The GO enrichment analysis showed that the nine HPGs were mainly related to cellular response to biotic stimulus, intrinsic apoptotic signaling pathway and response to lipopolysaccharide among the BP category terms. Among CC terms, the HPGs were significantly enriched in ruffle membrane, leading edge membrane and ruffle. Among MF terms, the HPGs were mainly enriched in cytokine receptor binding, cytokine activity and receptor ligand activity involved in the apoptotic process. Together, these findings showed that the nine HPGs are mainly enriched in immune-related signaling pathways (Fig. 9A). In addition, KEGG analysis showed that nine HPGs were mainly enriched in the NF-kappa B signaling pathway and inflammatory bowel disease (Fig. 9B).

Figure 9 Functional enrichment analysis of the nine HPGs.

(A) Bubble graph for GO enrichment. A larger bubble indicates a larger number of enriched genes, and redder color indicates more significant differences. (B) Bubble graph for KEGG pathways.

These findings indicated that the nine HPGs play important roles in immunity and participate in immune factor regulation.

Results of immune infiltration analysis

To further clarify the correlation between the nine HPGs in the model and immunity, as shown in Fig. 10, we further analyzed the association of the risk score model and immune cells. We found that the infiltration levels of B cells (Fig. 10A, Cor = 0.329 and P = 1.565e−13), CD8 + T cells (Fig. 10B, Cor = 0.407 and P = 1.851e−20), CD4 + T cells (Fig. 10C, Cor = 0.255 and P = 1.569e−08), Macrophages (Fig. 10D, Cor = 0.372 and P = 3.977e−17), Dendritic cells (Fig. 10E, Cor = 0.406 and P = 2.531e- 20) and Neutrophils (Fig. 10F, Cor = 0.412 and P = 6.28e−21) were positively correlated with risk values.

Figure 10 Identification of the correlation between the risk score and immune cell infiltration in LGG.

(A) B cells (Cor =0.329 and P = 1.565e −13). (B) CD8+ T cells (Cor =0.407 and P = 1.851e −20). (C) CD4+ T cells (Cor =0.255 and P = 1.569e −08). (D) Macrophages (Cor =0.372 and P = 3.977e −17). (E) Dendritic cells (Cor =0.406 and P = 2.531e −20). (F) Neutrophils (Cor =0.412 and P = 6.28e −21). Cor >0 indicates that immune cell infiltration is positively correlated with the risk score.

The results showed that the expression of immune cells in the high-risk group was increased relative to that in the low-risk group among LGG patients.

Discussion

At present, using risk scoring models to predict the prognosis of cancer patients and conducting clinical interventions as early as possible has become a research hotspot in various oncology disciplines (Gu et al., 2021; Zhao et al., 2021). However, in recent years, several studies have shown that pathological stage and methylation of the MGMT promoter are not sufficient to accurately predict the prognosis of glioma patients (Egaña et al., 2020; Weller & Reifenberger, 2020). Recent research showed that Chao et al. (2021) constructed a risk score model to predict the prognosis of glioma patients based on seven PGs identified as prognosis-related genes. However, the prognostic value and mechanism of PG activities in LGG remain unclear. Our research demonstrated that a risk scoring model constructed from PGs as newly identified biological targets accurately predicts LGG patient prognosis and verified the accuracy and stability of the model, which provides new options for personalized clinical treatment for LGG patients.

Pyroptosis, an important form of immunogenic cell death, was first discovered as a key mechanism against infection in 1992 in bacterially infected macrophages (Zychlinsky, Prevost & Sansonetti, 1992). Pyroptosis was originally thought to be a subtype of apoptosis (Wei et al., 2022). In 2001, however, Cookson showed that pyroptosis is a unique form of inflammatory programmed cell death dependent on caspase-1 (caspase-1) and gave it its unique name (Cookson & Brennan, 2001). Previous studies have shown that a variety of pathological factors (such as infection, malignant tumors, etc.) can trigger pyroptosis, which changes the permeability of the cell membrane. Holes are formed in the cell membrane, and Gasdermin family proteins mediate the entry of extracellular water into the cells, causing cell swelling and rupture, which ultimately promotes the inflammatory response through the release of inflammatory factors such as IL-1 β and IL18 (Yu et al., 2021). In recent years, with the continuous deepening of research, the relationship between pyroptosis and tumors has become increasingly apparent (Lu et al., 2022). To date, a number of pioneering results have shown that the promotion of tumor cell pyroptosis exerts antitumor effects in a variety of malignant tumors (digestive system Wu et al., 2019), respiratory system (Wang et al., 2018), blood system (Yu et al., 2019). In addition, with the development of precision medicine and personalized medicine in recent years, PGs have been widely used to predict the prognosis of cancer patients. Li et al. (2021) predicted the prognosis of GBM patients by constructing a scoring model including three HPGs. Subsequently, based on four HPGs, Lin et al. (2022) constructed a risk scoring model to predict the prognosis of GBM patients. In addition, Zeng et al. (2022) established an effective risk scoring model based on eight HPGs to analyze the prognosis of glioma patients. In summary, PGs are an effective biological target and will be used for the prognosis and personalized treatment of LGG patients.

Recently, based on an analysis of PGs, Zhou et al. (2021) constructed a scoring model through 6 HPGs, providing new insights for the treatment of LGG patients. In contrast to previous studies, we applied a new algorithm to obtain 45 DEPGs between normal brain tissue and tumor tissue through differential gene analysis and screened this set of genes for HPGs for building the scoring model. Ultimately, we obtained nine HPGs, specifically, CASP4, TP63, TIRAP, CASP9, IL18, TNF, IL1A, PLCG1, and TP53. In addition, different from the study of Zhou et al. GO and KEGG analyses of nine HPGs showed that immune-related genes were enriched. The results in Fig. 10 show that the abundance of immune-related cells was increased in the high-risk group increased, which also implies a correlation between the nine HPGs and immune-related genes.

Evidently, the development and prognosis of most cancers are related to immune regulation (Habif et al., 2019). Among HPGs, CASP4, an important participant in the pyroptosis pathway, plays a vital role in the release of immune factors (Moretti et al., 2022); moreover, it is closely related to the occurrence and development of multiple tumors (Flood et al., 2015; Yang et al., 2015). As a key protease and an important promoter of the mitochondrial apoptosis pathway, CASP9 was found to play an important role in the regulation of the immune microenvironment (Bratton & Salvesen, 2010). A recent study showed that the inhibition of CASP9 expression can mediate antitumor immunity, which benefits from the release of IFN-I in tumor cells after radiotherapy (Han et al., 2020). IL18, an important immune factor, is involved in various inflammatory responses and the regulation of tumor cells in the tumor microenvironment (Yasuda, Nakanishi & Tsutsui, 2019). A study by Markowitz et al. (2016) showed that IL18 can exert an inflammation-dependent tumor suppressor effect by promoting the differentiation, activity and survival of tumor-infiltrating T cells. IL1A, another immune factor, is involved in downstream signaling activated by the IL1R1 receptor (Olan & Narita, 2021) and thereby regulates a variety of tumor cells, such as cervical and colorectal cancer cells (Ji et al., 2019; Li et al., 2020). TNF is an important inflammatory factor that induces many cellular responses, including inflammation, cell proliferation, apoptosis and necrosis (Zhao et al., 2012). Recent evidence shows that it can promote tumor growth by recruiting neutrophils and macrophages in the tumor immune microenvironment (Salomon et al., 2018). TIRAP is a key intracellular signaling molecule that regulates a variety of immune responses and plays an important role in the immune response by interacting with a range of intracellular signaling mediators (Rajpoot et al., 2021), and some studies have shown that it also participates in tumor regulation. In non-small cell lung cancer (NSCLC) cells, downregulating TIRAP can significantly inhibit the proliferation of tumor cells (Hao et al., 2019). PLCG1, a member of the phospholipase C (PLC) family of enzymes, is involved in the development of glioma through the activation of growth factors such as EGFR and PDGFR (Walker et al., 2018). TP53 and TP63 are also involved in the regulation of tumor cell proliferation, migration and invasiveness (Mills et al., 1999; Yang et al., 1998) and immune regulation (Textor et al., 2011; Yang et al., 2011). Previous studies have shown that compared to oligodendrogliomas, TP53 as a gene with frequent mutations is more likely to mutate in astrocytomas. In addition, Research by Kim et al. (2010) show that TP53 mutations in patients with low-grade diffuse glioma are the cause of decreased survival rate. Therefore, the above studies corroborate our findings. However, how the nine HPGs identified here regulate immune factors in the high- and low-risk groups was not investigated in this study. This is a limitation of our study and an important area for further research.

Conclusions

In all, we developed a risk score model for LGG comprising nine HPGs, and the accuracy of the model was verified by Kaplan-Meier analysis of the training group and an independent test group. These nine HPGs will provide more gene targets for LGG patients. In addition, this study provides new opportunities for the individualized treatment of LGG patients, which has important clinical implications.

Supplemental Information

Supplemental Information 1 Construction of the risk score model by 45 DEPGs

(A) Univariate Cox regression analysis of OS for 29 prognostic genes. Five prognostic genes (HR <1) were protective genes, and 24 prognostic genes (HR¿1) were risk genes. (P < 0.05). (B) LASSO analysis was used to screen genes with high correlation. The horizontal axis indicates the logarithm of the independent variable lambda, and the vertical axis indicates the error of cross-validation. (C) Tuning parameter (λ ) selection cross-validation error curve for LASSO analysis.

Click here for additional data file.

Supplemental Information 2 The CNV status of 9 HPGs

(A) The CNV status of CASP4. (B) The CNV status of TP63. (C) The CNV status of TIRAP. (D) The CNV status of CASP9. (E) The CNV status of IL18. (F) The CNV status of TNF. (G) The CNV status of IL1A. (H) The CNV status of PLCG1. (I) The CNV status of TP53.

Click here for additional data file.

Supplemental Information 3 PCR

Click here for additional data file.

Additional Information and Declarations

Competing Interests

Author Contributions

Human Ethics

Data Availability

The authors declare there are no competing interests.

Hua Wang performed the experiments, prepared figures and/or tables, and approved the final draft.

Lin Yan performed the experiments, prepared figures and/or tables, and approved the final draft.

Lixiao Liu analyzed the data, prepared figures and/or tables, and approved the final draft.

Xianghe Lu analyzed the data, prepared figures and/or tables, and approved the final draft.

Yingyu Chen performed the experiments, prepared figures and/or tables, and approved the final draft.

Qian Zhang performed the experiments, prepared figures and/or tables, and approved the final draft.

Mengyu Chen performed the experiments, prepared figures and/or tables, and approved the final draft.

Lin Cai conceived and designed the experiments, authored or reviewed drafts of the article, and approved the final draft.

Zhang’an Dai conceived and designed the experiments, authored or reviewed drafts of the article, and approved the final draft.

The following information was supplied relating to ethical approvals (i.e., approving body and any reference numbers):

This study was approved by the Clinical Research Ethics Committee of the First Affiliated Hospital of Wenzhou Medical University (permission: 2016-076).

The following information was supplied regarding data availability:

The data is available at TCGA (TCGA-GBM from Brain), GTEx (GTEx Analysis V8. 3. in the GTEx Portal) and GEO (GSE43378).

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
