# Peer review of "A pyroptosis gene-based prognostic model for predicting survival in low-grade glioma"

_PeerJ, doi:10.7717/peerj.16412_

## Round 0.1 · original submission · Major Revisions

Please address questions and comments raised by reviewers and resubmit.

Reviewer 1 ·

Basic reporting

The text is well written and all the references and figures are properly cited in the text. There are minor formatting errors which can be rectified. A thorough read by a professional English speaker will help improve on the text.
Figures are well represented. Addition of statistical details in graphs and or in the adjoining captions is recommended to make the figures more independent and informative. I recommend integration of 4 and 5 as they address the same topic and if represented together then would be more informative to the reader.
Figure 8C, the curve is seen to be altered when classified for age, gender and grade. Please reconfirm with data and interpretation.
Figure 2B, it will be more helpful if names of the genes are mentioned in the plot. At least those which show higher differential expression.
Under immune infiltration analysis, is there a higher expression of gene set in immune cells or there is a predominance of the immune cells in the tumor micro environment?

Experimental design

I appreciate the authors approach to describe the study model at the start.

Validity of the findings

Perhaps an iteration of the analytical model on multiple nonrelated diverse datasets would help stratify the model system.

Additional comments

NO.

Reviewer 2 ·

Basic reporting

Review
A pyroptosis gene-based prognostic model for predicting
survival in low-grade glioma

by Hua Wang, Lin Yan Equal first author, , Lixiao Liu , Xianghe Lu , Yingyu Chen , Qian Zhang , Mengyu Chen , Lin Cai , Zhangan Dai

analysed LGG tumours for their differential gene expression of pyroptosis genes with respect to healthy brain. They identified 9 hub genes which were used to divide tumours into a high and a low risk group.

Major:
Methods reproducibility and availability:
In principle I support the bioinformatics (scoring) approach applied by the authors. I personally reviewed in the last years at minimum four papers using this method in an almost identical fashion (using different functional aspects and cancer types) as I see based on the fully similar figure types. The methodical steps can be reproduced only in a very fragmented manner by interested re-users.

Its time to make this method available, as a pipeline, e.g. at GitHub for bioinformatics community for reproducing the analysis.

LGG contextualization
The paper focuses on pyroptosis and related genes, what is justified to some degree but also an arbitrary choice. One can choose dozens of other functions to construct analogous scores with (probably) similar differences between high- and low- risk groups. For a broader understanding I miss completely how the pyroptosis-centered results fit into generally accepted classification schemes of LGG, e.g. by WHO, namely using genetic markers: mutations of the IDH gene and chromosomal deletions on Chromosome 1 and 19 that divide LGG into three major groups with associations to histo-characteristics. Moreover, additional mutations, eg of the TP53 gene (part of the score) strongly associate with these groups as well as immune-functions discussed by the authors. Moreover, these genetic markers strongly affect prognosis of LGG.
I suspect that the pyroptosis score simply associates with those genetic groups, i.e. that wildtype IDH and mutated IDH mutated plus intact Chr1 and 19 cases strongly enrich in the high risk group while Chr1 and 19 codeleted cases enrich in the low risk group.
The authors have to consider the genetic status of the LGG accordingly. This should include also a fair comparison with their " What else?

Line 213-215: “ The interactions among the 45 DEPGs are shown in Figure
2C. Specifically, excluding GSDMB and GSDMC, which shared a unique interaction, the
remaining 43 DEPGs all exhibited multiple interactions.” Figure 2C is not understandable. The sentence doesn’t make sense to me.

Line 228- 232: Please reduce the number of digits of the coefficients (only significant digits should be shown).

Line 253 ff: Does the sign of differential expression agree with the signs of the prediction model? Please indicate.

Line 256ff: The change differential expression (decay in low risk group) refers throughout to genes upregulated in astrocytomas (compared with oligodendrogliomas) what supports my suggestion according to association with the genetics (see above), namely that astrocytomas have poorer prognosis than oligodendrogliomas. The authors re-tell a well known story with their genes. Moreover, one of the marker genes locates at Chr 1 and there seems to be an accumulation of other markers at Chr. 11 …please discuss and check…

Line 262: TP53 is frequently mutated in astrocytoma-like LGG what is in-line with my rationale above. Again: The authors reproduce well known facts without mentioning this. The mutation rate of the other genes seems to be insignificant. Please check their CNV status.

289ff: Please compare survival statistics of the risk groups with that of WHO genetic groups. Do you find improvement.

307ff: What means G2 and G3 groups. Why sex and age are pathological classes. Theses phrases are not clear.

317ff: The authors find enrichment of immune and inflammatory functions at high risk and report different signatures of infectious diseases (Salmonella etc). These are obviously secondary agreements, has nothing to do with brain cancer and can mislead the reader. It needs cleaning.

Line 345: MGMT discussion is outdated and not related to the results of the paper?

Line 385ff until end of results: This is gene poetry and should be more directly related to known LGG genomics/transcriptomics.

Fig 2B: Please assign the relevant genes in the volcano plot.

Fig 2C: Seems without relevance?

Fig 3: Can be shown as supplement. Its technically related to the method.

Fig 4: Please indicate gene-symbols in the figure, like in Fig 5.

Fig 6: What do the plusses (+) mean and how do they relate to SNV, CNA?

Fig 7: There is large difference between training and test data what makes independent check and comparison based on genetic grouping necessary.

Fig 11: Probably I overlooked, how immune cell infiltration was determined? What is timer ?

I suggest reducing the number of figures in the main paper and shift 2-3 into the supplement…

Experimental design

Standard, not novel (including bioinformatics).

Validity of the findings

Okay, but largely decoupled form state of the art knowledge on LGG...Needs considerable improvement...

Additional comments

Overall focussing on one functional mechanism (pyrptosis) but neglecting overall knowledge is a serious flaw of the paper. On the other hand, pyroptosis-scientists will enjoy, probably.

---

## Round 0.2 · Minor Revisions

We are returning the manuscript to you for further editing. We have noticed that several relevant literature references are missing. Please see below just a few examples:

Lin J, Lai X, Liu X, Yan H, Wu C. Pyroptosis in glioblastoma: A crucial regulator of the tumour immune microenvironment and a predictor of prognosis. J Cell Mol Med. 2022 Mar;26(5):1579-1593.

Zeng Y, Cai Y, Chai P, Mao Y, Chen Y, Wang L, Zeng K, Zhan Z, Xie Y, Li C, Zhan H, Zhao L, Chen X, Zhu X, Liu Y, Chen M, Song Y, Zhou A. Optimization of cancer immunotherapy through pyroptosis: A pyroptosis-related signature predicts survival benefit and potential synergy for immunotherapy in glioma. Front Immunol. 2022 Aug 3;13:961933.

Yang et al., Front. Oncol., 31 March 2022
Wei, X., Xie, F., Zhou, X. et al. Role of pyroptosis in inflammation and cancer. Cell Mol Immunol 19, 971–992 (2022). https://doi.org/10.1038/s41423-022-00905-x

Lu, L., Zhang, Y., Tan, X. et al. Emerging mechanisms of pyroptosis and its therapeutic strategy in cancer. Cell Death Discov. 8, 338 (2022). https://doi.org/10.1038/s41420-022-01101-6

Zhang, M., Cheng, Y., Xue, Z. et al. A novel pyroptosis-related gene signature predicts the prognosis of glioma through immune infiltration. BMC Cancer 21, 1311 (2021). https://doi.org/10.1186/s12885-021-09046-2
Chen et al., 2023, Volume12, Pages 5071-5087

1. Please go through publications on this topic and cite relevant literature

2. Include details from relevant literature in the introduction section, methodologies section, and in discussion as necessary

3. For several published studies that are very similar to yours, include a paragraph explaining key findings from those studies, limitations and contributions of your work in light of those papers.

Reviewer 2 ·

Basic reporting

The authors addressed all my issues raised in the first review round. I dont have objections against publication.

Experimental design

See my comments in the first review round.

Validity of the findings

See my comments in the first review round.

Additional comments

no additional comments

---

## Round 0.3 · accepted · Accept

Congratulations! Your manuscript has been accepted for publication.

Reviewer 1 ·

Basic reporting

Please give a thorough read before publishing for subtle writing errors.

Experimental design

no comment

Validity of the findings

no comment

Additional comments

no comment